# Impact of Green Space Exposure on Children’s and Adolescents’ Mental Health: A Systematic Review

**DOI:** 10.3390/ijerph15122668

**Published:** 2018-11-27

**Authors:** Gert-Jan Vanaken, Marina Danckaerts

**Affiliations:** Department of Child and Adolescent Psychiatry, University Psychiatric Centre Leuven, KU Leuven, 3000 Leuven, Belgium

**Keywords:** green space, mental health, mental well-being, children, adolescents, urban planning

## Abstract

In recent years, the interest in the relationship between exposure to green spaces and children’s and adolescents’ mental health has risen. This systematic review aims to provide an overview of observational studies assessing the association between empirical green space exposure with standardized outcome measures of mental health problems, mental well-being and developmental problems in children, adolescents and young adults. The PRISMA statement guidelines for reporting systematic reviews were followed. A PubMed and Scopus search resulted in the inclusion of 21 studies. The evidence consistently suggests a beneficial association between green space exposure and children’s emotional and behavioral difficulties, particularly with hyperactivity and inattention problems. Limited evidence suggests a beneficial association with mental well-being in children and depressive symptoms in adolescents and young adults. These beneficial associations are resistant to adjustment for demographic and socio-economic confounders, which thus may represent independent links. Mediating factors and the variability of this association between different age groups are discussed. From a precautionary principle, evidence up to now demands the attention of policy makers, urban planners and mental healthcare workers in order to protect children’s and adolescents’ mental health in light of rapid global urbanization by providing sufficient exposure to green spaces.

## 1. Introduction

In recent years, academic interest in a possible relationship between exposure to green spaces and public health has risen, as can be derived from the number of studies published. In a systematic review of 12 reviews published between 2010 and 2016, the authors underscored the beneficial association of exposure to green spaces with all-cause mortality, mortality by cardiovascular disease and mental health in adults [1]. However only one included review, published in 2015 by Gascon et al. assessed mental health in children, finding inadequate evidence for such an association [2]. This incremental interest can be understood firstly, clinically in the light of the rapid global urbanization dynamic, triggering concerns of a decline of contact with natural green spaces [3,4,5], and secondly, methodologically in the availability of new remote sensing techniques to quantify green spaces using satellite images [6]. Of course urbanized regions can offer contact with green spaces as well, defined in this context as urban vegetated spaces such as parks, grasslands, cemeteries, sports and playing fields and near-road trees [7]. Following these evolutions, the importance of green spaces has also attracted political interest over the past years. In the United Nation’s Sustainable Development Goals for instance, member states committed themselves to the following goal, “By 2030, provide universal access to safe, inclusive and accessible, green and public spaces, in particular for women and children, older persons and persons with disabilities” [8].

Although green spaces seem to be associated with a range of health benefits, particular interest has arisen for the impact on children’s mental health. One of the main challenges in this field of research is disentangling the independent role of green space exposure from other confounding variables. Intuitively most attention was paid to the role of socio-economic status, as this factor predicts mental health and may influence where families eventually end up living. In 2009, Maas and colleagues published a landmark study linking data of residential land use to medical diagnoses in primary care settings for approximately 345,000 Dutch patients from different age groups [9]. Patients living in the greenest environments were significantly less frequently diagnosed with a range of physical and mental disorders after adjustment for the most probable socio-economic and demographic confounders, compared to their counterparts living in the least green neighborhoods. Remarkably, the strongest associations were found for psychiatric conditions, particularly in children [9]. Since then, a growing number of studies have assessed those associations between green space and children’s and adolescents’ mental health. As mentioned earlier, Gascon et al. conducted the first systematic review on this specific topic, in 2015, considering the results of the four existing observational studies as inadequate evidence at the time for a causal relationship between green spaces and children’s mental health [2]. More recently, in 2017, McCormick published another systematic review in this domain [10]. With a broader scope on children’s mental well-being and school functioning, including 12 articles in the review, the author concluded that access to green space was associated with improved mental well-being and cognitive development of children. However, it should be noted that heterogeneity in study design was significant and sufficient replication research was not systematically available at the time of review.

As global urbanization progresses, more than ever, children, adolescents and young adults are coming to age in neighborhoods dominated by built environments. At the same time, mental health problems in these age groups put a significant burden on these individuals and their life course, their families and society as a whole. Since the research line that sheds a light on the interplay of these dynamics expanded rapidly in recent years, there is a need for a new overview of the existing evidence to guide urban planning and public health policy. This paper aims to provide such an attempt at answering the question if an association between green space exposure and children’s and adolescent’s mental health and neurocognitive development exists and which domains of these outcome concepts are possibly involved in particular.

## 2. Materials and Methods

### 2.1. Search Strategy

This systematic review was conducted following the Preferred Reporting Items for Systematic reviews and Meta-Analyses (PRISMA) statement guidelines [11]. The literature search was carried out using two frequently used search engines, PubMed and Scopus. Following an exploratory phase to identify the essential aspects in this domain of research, the following keywords were selected: Keywords related to green space (“green space”, greenspace, greenness) combined with keywords related to mental health and neurocognitive development (“mental health”, “well-being”, development *) and with keywords related to childhood and adolescence (child *, adolescen *, infan *, “young adult *”, youth). The asterisk (*) acts as a truncation symbol to detect various results composed of a single string of text, e.g. ‘child *’ detects terms as ‘children’ and ‘childhood’. On 2 March 2018, the last search was conducted. The title and abstract provided the information for a first screening of the given articles. Following this first screening, full articles were read to decide on inclusion for this systematic review. References of the included studies were taken into account to identify additional articles.

### 2.2. Selection Criteria

The selection criteria for inclusion that all studies needed to fulfill were: (a) The article described an original research study; (b) the article was written in English; (c) the article was published in a peer-reviewed scientific journal; (d) the study had an observational design, interventional designs were excluded; (e) green space exposure was analyzed empirically either by objective methods based upon land cover maps or remote sensing data, or by subjective methods using standardized questionnaires; (f) green space exposure was assessed in residential or school surroundings; (g) mental health and neurocognitive development were assessed using standardized instruments or through existing medical records or databases; (h) the study focused primarily on children, adolescents or young adults with an age limit of 25 years, articles primarily focusing on adults over 25 years of age and were excluded even when the study population existed in part of participants under the age of 25 and (i) no restriction on publication date was put in place.

### 2.3. Data Collection and Evaluation of Evidence

Following the methodology used in a previous systematic review on green space and mental health by Gascon et al., [2], we extracted the ensuing basic characteristics, methodological features and outcome results of the studies included in our systematic review: Author, year of publication, country, study design, study population, sample size, exposure assessment, outcome assessment, confounding factors and main results. These extraction processes were conducted manually.

As children’s mental health and development are broad concepts, with a multitude of existing methods to capture these concepts, the available evidence for any impact of exposure to green spaces was evaluated based on the studied outcome parameter. These outcome parameters were divided into three categories, i.e., (1) emotional and behavioral difficulties; (2) mental well-being and (3) neurocognitive development. The evaluation of evidence for an association between green space exposure and any studied outcome parameter was based on the number of studies finding a beneficial association in comparison to the total number of studies assessing this parameter.

## 3. Results

### 3.1. Literature Search Results

The literature search using the described keywords yielded 89 results in PubMed and 197 results in Scopus, as presented in Figure 1. A first screening based on title and abstract resulted in 48 articles after the elimination of duplicates. After full-text read-through, 21 studies were selected for inclusion. No additional articles fitting the predefined criteria were detected via the references.

### 3.2. Study Characteristics and Methods

As presented in Table 1, more than half of the included studies were published in 2016 or later (*n* = 12), while the earliest publication matching the inclusion criteria only dates back to 2013. Eleven of the selected studies were conducted in Europe, others were conducted in North America (*n* = 6) and Oceania (*n* = 4). Twelve studies had a cross-sectional design, seven studies were longitudinal and two studies were ecologically designed, i.e., relying on exposure and outcome data at the district instead of the individual level [12,13]. The sample size of the examined populations ranged between 72 [14] and approximately 3,000,000 participants [12]. Exposure to green space was measured in the majority of studies with at least one objective parameter. Land cover maps providing data on land use (*n* = 10) or remote sensing data based on satellite images providing the Normalized Difference Vegetation Index (NDVI) (*n* = 9) and derivations, were the most commonly used objective green space parameters. The NDVI is a parameter based on the difference in reflected visible and infrared light between green spaces and the built environment. One study used geolocation data obtained by study participants wearing digital watches with GPS-functionality [14]. Two studies assessed green space exposure solely with a subjective parameter, i.e., the score on a questionnaire gauging time spent in green spaces or the distance to the nearest green space [15,16]. Seven studies combined objective and subjective parameters in the exposure assessment.

The majority of studies (*n* = 16) assessed green space exposure only in the residential neighborhood of children or adolescents, two studies measured exposure exclusively in the school environment while three studies combined residential and school exposure to green spaces.

Mental health outcome was assessed by a range of different instruments. Eight studies used the parent-based version of the Strengths and Difficulties Questionnaire (SDQ), while one study relied on the SDQ filled out by multiple informants [17]. Eight studies relied on other standardized questionnaires. Computerized neuropsychological tests were used to assess aspects of neurocognitive development in three studies. Medical records and derived prevalence data provided an outcome measure in two studies. Outcome results were described heterogeneously as the percentile change in average score on a given test, the odds ratio, the rate ratio or the beta coefficient of correlation. Associations that are presented in Table 1 or described below always consist of significant and confounder-adjusted associations, unless stated otherwise. All studies assessed at least one measure of the socio-economic status (SES), i.e., family income, parental education and/or parental employment, except for two studies [14,15]. Adjustments for ethnicity, parental mental health and neighborhood socio-economic status (nSES), i.e., safety, walkability, house value, average household income of the neighborhood, were conducted less systematically, as presented in Table 1.

### 3.3. Evaluation of Evidence

#### 3.3.1. Emotional and Behavioral Difficulties

As mentioned earlier, nine studies evaluated emotional and behavioral difficulties in children using the internationally validated Strengths and Difficulties Questionnaire (SDQ), giving opportunities for the comparison of respective results. All but one study relied on the parent-rated version, although a teacher-rated version and a self-rated variant suitable from age eleven, were equally available. The SDQ consisted of five domains, i.e., emotional symptoms, peer problems, hyperactivity and inattention symptoms, conduct problems and prosocial behavior. The first four domains could be summed up resulting in the Total Difficulties Score (TDS), a measure of general mental health. Additionally, an internalizing (emotional symptoms and peer problems) and externalizing subscale (hyperactivity and inattention symptoms and conduct problems) could be derived. However, not all studies reported separately on these different domains or subscales. Studies differed substantially in design, study population and assessed exposure parameters as presented in Table 1. This group of studies described pre-school and school-aged children between the ages of 3 and 13. Seven out of those nine studies reported on the Total Difficulties Score after adjustment for demographic and socio-economic confounders. All of those seven studies documented a significant confounder-adjusted association between a measure of green space exposure and the Total Difficulties Score [16,17,18,19,25,29,30]. Adjusting for the assessed confounders did not systematically shift the results in one direction, i.e., it depended on the specific city or country context whether taking into account demographic and socio-economic confounders would strengthen or weaken the associations. Quantitative comparison of the size of association was limited because of different exposure mapping and variation in measures of presentation of the results, as mentioned earlier. However, where sufficient data were available, intra-study comparison highlighted that confounder-adjusted associations between green space exposure and the Total Difficulties Score were comparable in effect size to the associations of the family socio-economic status (SES) with this same outcome measure [16,17,25,30].

Regarding the different domains of the SDQ, the strongest results were found for the hyperactivity and inattention domain. Five out of six studies documented significant associations with hyperactivity and inattention problems [16,18,19,29,30]. Smaller associations were found in four out of five studies for peer problems [18,19,29,30] and less consistently in two out of five studies for conduct [19,30] and emotional problems [18,26]. Richardson and colleagues [30] found a modest correlation between prosocial behavior with the amount of residential public green space as compared to private garden access, being the only study assessing this domain in the light of a different exposure between public and private green spaces. Two other studies did not find associations between prosocial behavior and measures of residential green space [18,19].

Next to the Total Difficulties Score, Feng et al. [17] and Feng and Astell-Burt [25] presented scores, of 4 to 13 year old children, on the internalizing and externalizing subscale of the SDQ, being the only two studies to do so. Green space quality and quantity were associated with both of the parent-rated subscales scores for all age groups. For the oldest group in this population, i.e., age 12 to 13 years, the associations with the internalizing subscale were tested and confirmed on the self-rated version of the SDQ. Longitudinal results showed that associations with green space quality grew notably stronger compared to green space quantity, in this older age group of the study population. Green space quality was defined subjectively in this study as the parents’ answer on the question to what extent good parks, playgrounds and spaces to play were available in their neighborhood.

Studies that analyzed different types of green space exposure showed that a shorter distance to the nearest green space and access to a private garden were stronger associated with the SDQ than the average amount of green space in the neighborhood [16,19,29,30]. Although beneficial correlations to the SDQ scores were described for all studied age groups, longitudinal studies are not equivocally presenting an impact of green space on the trajectory of these scores [25,26,30].

In a large, prospective twins birth cohort study, Younan et al. [32], found short—(one to three months) and long-term (one to three years) exposure to neighborhood green space to be associated with reduced aggressive behavior in Californian children and adolescents, as reported by parents on the aggression subscale of the Child Behavior Checklist (CBCL). The authors described a decrease in the CBCL-Aggression score of 0.4 points for an average of 4.86 (SD 5.03) when increasing surrounding greenness over an interquartile range, i.e., the difference in exposure between the lowest versus highest quartile of the population. When compared to the decline in aggressive behavior over time seen during normal development, the decrease in aggression scores associated with higher levels of exposure to green space was equivalent to approximately two years of behavioral maturation.

In an American cohort of adolescents, Bezold and colleagues [20] documented an 11% decrease in depressive symptoms on the self-rated McKnight Risk Factor Survey in relationship to an interquartile range increase in residential neighborhood greenness. In a following study, partially based on the same data, the authors found the cumulative greenness exposure during childhood and adolescence to be associated with a 6% decrease in the incidence of high depressive symptoms in adolescents and young adults, with even stronger associations in highly urbanized neighborhoods [21].

No direct association between indicators of green space exposure and mental health as measured on the self-reported 12-item General Health Questionnaire was found by Dzhambov and colleagues [24] in their study based on a sample of 399 15- to 25-year olds. However, the authors detected indirect associations of green space exposure with mental health via serial mediation of physical health, neighborhood social cohesion and perceived restorative quality, a “measure of psychologically restorative qualities of environmental experience grounded in attention restoration theory” [24].

#### 3.3.2. Mental Well-Being

In the city of Houston, USA, Kim et al. [28] found that for 9- to 11-year old children, more and larger urban green spaces in the neighborhood were positively associated with their health-related quality of life, as reported by themselves and their mothers, on the Pediatric Quality of Life Inventory (PedsQL). Ward et al. [14] collected objective data on 72 adolescents on time spent in green spaces and physical activity relying on data from GPS-facilitated sports watches. Both exposure measures showed robust associations with adolescents’ self-rated life satisfaction, well-being and happiness, but not with their results on a computerized battery of neurocognitive tests. Time spent in green spaces and physical activity proved to be partially interdependent but were also both independently related to increased scores on the emotional well-being scales, with green space exposure having the strongest relationships. It should be noted however, that this study was the only one that did not include a measure of SES as a possible confounder in the statistical adjustment model. Huynh and colleagues [27] did not find any significant associations for the average greenness in a 5000 m buffer surrounding Canadian adolescents’ schools with self-reported emotional well-being on the Cantrill ladder. Neither were Saw and colleagues [31] able to show that access or use of green spaces affected emotional well-being of older adolescents and young adults in the tropical city of Singapore.

#### 3.3.3. Neurocognitive Development

In the city of Berlin, Germany, Kabisch and colleagues [13] analyzed the results of community-organized medical check-ups of pre-school in children in relation to green space exposure on a subdistrict level. The authors found an inverse correlation between the percentage of naturally covered land and deficits in visuo-motor and, to a lesser extent, language development. Dadvand and colleagues [22] documented an enhanced twelve-month developmental progress in working memory and superior working memory, and a reduction in inattentiveness in primary school children that were exposed to higher levels of surrounding greenness. The authors relied on repeated computerized *n*-back tests (respectively two- and three-back) and the *Attention Network Task* (variability of reaction time). Results on these tests differed by +5%, +6% and −1%, respectively per interquartile range change in total, i.e., school-based and residential, greenness exposure. Concentrations of elemental carbon, a traffic-related air pollutant were assessed and explained 20–65% of the variances in neurocognitive outcome. No association was found when only residential greenness was considered in contrast to the school-based or the combined exposure. In a following study, comprising a younger population of age four to seven, Dadvand et al. [23] found an association between specifically greater lifelong residential green space exposure and enhanced development of sustained attention, derived from the results of the computerized *Attention Network Task* and *Continuous Performance Task*.

Within the pool of identified articles, only one study looked at a possible relationship betwen green space exposure and the prevalence of autism. Wu et al. [12] conducted an ecological study in the state of California, United States, finding negative correlations between different types of green spaces on elementary school district-level, i.e., forests, grasslands and (near road) tree canopy, and autism prevalence, derived from an existing prevalence database. In stratified analyses based on the road density of the school districts, these correlations remained intact in the quartile of districts with the highest road density, but they did not in other quartiles. In these districts with the highest road density, the authors described for every 10% increase in forest, grassland, average tree canopy and near-road tree canopy, a decrease in autism prevalence of respectively 10%, 10%, 11% and 19%.

## 4. Discussion

### 4.1. Evidence for an Association between Green Space Exposure and Mental Health

The aim of this review was to provide an overview of observational studies assessing the association between green space exposure and a variety of mental health outcomes in children and adolescents. The seven studies reviewed regarding emotional and behavioral difficulties, present credible evidence for an association between green space exposure and the Total Difficulties Score of the Strengths and Difficulties Questionnaire, a measure of general mental health in children. Regarding the subdomains that compose this general score, there is considerable evidence for a consistent and significant correlation between green space exposure and hyperactivity and inattention problems, suggested by four out of five studies assessing this subdomain. This finding is supported by two longitudinal studies demonstrating the association between green space exposure and the neuropsychological development of attention and working memory, relying on computer-based tests. Furthermore, both studies that investigated the parent- and self-rated emotional well-being and quality of life of children in the light of green space exposure did find a beneficial correlation. Less research is available for emotional and behavioral difficulties in adolescents. However, two studies that assessed self-rated depressive symptoms in this age group, did find an inverse correlation with green space exposure. One study that investigated aggressive behavior in adolescents in a longitudinal design, also found a significant inverse relationship. Emotional well-being in adolescents was assessed in one study without finding a beneficial association. The authors of this early study, published in 2013, assessed the effects of green spaces in a rather large exposure, i.e., 5000 m around the adolescent’s schools [27]. This may possibly explain the absence of finding a clear association. Later studies systematically assessed smaller exposure ranges. Two studies in this review assessed mental well-being in young adults, however, both did not find a beneficial association with green space exposure. A small sample size [24] and a tropical climatological setting [31] could explain these findings. It was noted that young adults were hardly defined as a separate population, explaining why a small number of studies focusing on 18- to 25-year olds, were identified using the predefined inclusion criteria.

### 4.2. Independency of the Association

These described relationships with mental health outcomes are all associations that were adjusted for demographic (age, gender, ethnicity) and socio-economic confounders (parental education, income, employment), as those are suggested predictors of mental health. Results on the SDQ and other outcome parameters after adjustment for these confounders were not systematically shifted towards stronger, weaker or similar associations. This can be understood by the fact that in some cities or countries green space is spread out equally between the strata of socio-economic status or ethnical background, such as in the city of Barcelona, Spain [18]. In other settings, inequalities do exist, e.g., in Scotland the socio-economically most deprived children had lower odds of having private garden access or a public park in their neighborhood [30]. On the other hand, in the city of Kaunas, Lithuania, families of higher socio-economic status lived further away from green spaces than their counterparts of lower socio-economic status [19]. This suggests that depending on the city or country of interest, these confounding variables work out differently. As mentioned earlier, confounding by ethnicity, neighborhood socio-economic status and parental mental health were assessed less systematically. However, it could be noted that associations in studies that adjusted for these concepts did still find significant associations in line with studies with a more modest adjustment methodology. Therefore, based on these results, it can be presumed that there is strong evidence for an association between green space exposure and mental health in children and adolescents, independent of the aforementioned confounders.

### 4.3. Direction of the Association

Although our main hypothesis was that green space exposure causes better mental health, it could be hypothesized that the described association between green spaces and mental health works out in the opposite direction of what is generally presumed. This alternative, inverse hypothesis would imply a mechanism that makes parents of a child with more mental health problems, more likely to reside in a less green neighborhood. Longitudinal studies such as those by Dadvand et al. showed that from birth to age 7, about 25 % of the children changed addresses, a significant percentage [23]. A clear rationale to explain a theoretical mechanism is lacking, but since none of the included longitudinal studies presented data on a possible association between children’s mental health problems and their odds of moving towards less green neighborhoods, this possibility cannot yet be dismissed. Another theoretical explanation for this alternative hypothesis could imply a gene-environment interaction making parents with hyperactivity and attention deficits more likely to choose more stimulating neighborhoods and thus possibly less green environments to live in. If true, this would mean that finding a greater share of children with hyperactivity and inattention problems in less green neighborhoods is explained by a genetic rather than an environmental risk factor. However, based on the presented evidence, parental mental health does not seem to attenuate the documented associations fully, making this hypothesis less plausible. Other contra-arguments regarding this alternative hypothesis can be found, as we discuss below, in the evidence for mediating mechanisms that explain an association in the direction from more green space exposure towards better mental health. Nonetheless, it would be valuable for future research to assess the alternative hypothesis in depth by adjusting more profoundly for parental mental health and specifically for relevant domains such as hyperactivity and inattention problems. Additionally, in longitudinal research, data could be collected on children’s mental health problems and their odds of moving towards less green environments.

### 4.4. Mediation of the Association

In previous work, a multitude of mediators of the association between green space and mental health were suggested [2]. The included studies in this review, provide some evidence for the roles of air pollution, physical activity and social interaction, adding plausibility to the causal role of green space exposure. Firstly, the protective role of green spaces against air pollution was especially examined for neurocognitive development outcome parameters such as (superior) working memory, inattentiveness and autism prevalence [12,22]. Although this evidence is limited, it adds consistency to the growing evidence for the concerning, detrimental role of air pollutants on children’s brain development, increasing the odds of having an impaired attentional function and autism spectrum disorder [33,34,35]. Research in adult populations also suspects that exposure to poor air quality increases the odds of depression [36]. Bezold et al. found no such role for fine particulate matter as a mediator in the association between green space exposure and depressive symptoms in adolescents [21]. In this study, however, average levels of this subgroup of air pollutants hardly differed between the strata of green space exposure, possibly explaining the negative findings. Buffering of noise annoyance, which often correlates with traffic load and indirectly with air pollution, was not assessed separately as a mediating mechanism in the included studies. However, this factor is known to be associated with behavioral problems in children [37] and thus could be another plausible mediator that future research could investigate.

Secondly, Ward et al. provided objective data on time spent in green spaces and physical activity relying on data by GPS-facilitated sports watches. Spending time in green spaces was related to higher levels of physical activity and to better mental health. Associations of green space exposure with children’s well-being were in part attenuated by separately adjusting for their physical activity in those green spaces, suggesting a partial mediating role for physical activity [14]. Lastly, some evidence is available for the role of social interaction as another plausible mediator of the association. Green space playing time, in contrast to the mere availability of green spaces [18], was strongly associated to less peer problems, and access to public instead of private green space was modestly associated with increased prosocial behavior [30].

Since evidence is available for the beneficial effects of green spaces on adults’ mental health [1], parental mental health is a tentative mediating factor as well. Some studies adjusted for possible confounding effects of parental mental health, however in the included studies, mediation analysis of this factor was lacking.

### 4.5. Variability by Age Group

Green space exposure seems to be correlated with children’s and adolescents’ mental health in different ways depending on their developmental level. In children, especially, the availability of a private garden and a shorter distance from their residential address to a green space seems to be beneficial [15,19,30]. This observation may reflect the smaller environmental circle that is accessible at this age. For older children and adolescents, associations with mental health are found more consistently with average greenness in the surrounding neighborhood and with the quality of green spaces, suggesting that this age group benefits not merely from the closest green space, but rather from green spaces of their choice, reflecting their growing autonomy [25].

### 4.6. Relevance of the Association

In general, concluding a causal pathway between green space exposure and children’s and adolescents’ mental health might be preliminary. However, the presented associations are resistant to adjustment for the most important demographic and socio-economic confounders and there is considerable evidence for plausible mediators between exposure and outcome. Therefore, these results warrant, from a precautionary principle, attention from policy makers, urban planners and health care providers. The importance of green space exposure for public mental health is indicated by Younan et al. and Dadvand et al. Based on their data, they suggest that if surrounding greenness would be increased over an interquartile range, then respectively 12% and almost 9% percent of children in a clinical range of aggressive behavior and superior working memory impairment, would move out of this category [22,32]. This represents the fact that although individual effects can be rather small and do not explain the majority of variance in mental health problems, the effects are apparent across the full dimension of urban-dwelling youth and not only in the clinical range. Therefore, the effects of green spaces are of societal importance.

### 4.7. Strengths and Limitations

The strengths of this systematic review lie in the number of identified articles providing opportunities to compare results of similarly designed studies. Additionally, this review could include some longitudinal studies, studies on adolescent populations and studies relying on objective neuropsychological tests, which is a fairly recent evolution in this field. In contrast to the recent systematic review of McCormick [10], this review differed in search terms and inclusion criteria, focusing on observational studies and more prominently on reliable mental health and developmental outcome parameters. In comparison to the earlier systematic review on the topic by Gascon et al. [2], published in 2015, this review relied on a largely similar methodology, but could include 17 new studies examining the impact of green spaces on children’s and adolescents’ mental health, proving the emerging interest in this field of research. In addition to previous work, this review looked at green space exposure beyond a residential environment, including the school environment and focused in particular on the available evidence for different outcome domains. It can be noted as a strength that beneficial associations of green space exposure with mental health are presented, independent of the measurement instrument, i.e., multiple behavioral and quality of life questionnaires and neuropsychological tests, and independent of the information source, i.e., parents and children.

Limitations can be identified at the study-level, noting that the assessment of green space exposure differed largely between studies, although this difference also provided opportunities to assess which green space exposure concepts might be beneficial in particular. Additionally, the comparison of outcome results for high and low exposure groups were generally conducted within a given geographical region, making it difficult to define an optimal, absolute amount of green space exposure. Furthermore, it can be noted that mental health problems in adolescents such as addiction, that are not easily detected by general questionnaires as the SDQ or CBCL, fell out of the scope of this review.

At review-level, despite the thorough exploration of relevant keywords prior to the final search, an incomplete retrieval of results relying on the selected search terms and databases cannot be excluded. However, no additional studies were detected via the references of the search results and all studies matching the inclusion criteria mentioned in earlier systematic reviews were identified, suggesting that the conducted search was fairly adequate. Grey literature, i.e., publications by organizations or governments published outside of the traditional academic journals, was not systematically searched. Although urbanization occurs most quickly in terms of geography—in Africa and Asia—, and —in terms of economics—, in low- and middle-income countries [38], none but one study collected data in these settings.

### 4.8. Future Research Suggestions

Based on our discussion and the aforementioned limitations, it would be valuable if future research would assess differential effects of varying green space exposure concepts (i.e., time spent in, proximity and quality of green spaces etc.) and different exposure ranges simultaneously, taking into account non-residential exposure such as exposure in school and leisure environments. This future research could be particularly valuable in guiding policy-making and urban planning more concretely. Analogously, future reviewers of literature might benefit from using a broader set of keywords concerning non-residential green spaces. Keywords such as ‘natural environment’, ‘natural playground’, ‘green schoolyard’ etc., were up to now used to a rather limited extent within the field. However, these keywords might gain traction in the coming years as can be suspected from publications that appeared after the final search for this review had been conducted [39,40]. For reasons of comparability, use of the same outcome measures as in earlier research, such as the SDQ, is recommended. This instrument can be used for adolescent populations as well, since the questionnaire is validated for individuals up to 17 years old. The parent-based CBCL or the Youth Self Report are other valuable alternatives, since they provide broader and more clinical information. To overcome biases introduced by single informant questionnaires, the use of multi-informant variants are recommended. Research that addresses mental health problems outside the scope of the frequently used questionnaires would be welcomed. Gaming and internet addiction could be such a topic of interest since it can be hypothesized that screen time and green space exposure are inversely correlated [15] and since horticultural therapy has been mentioned as a relevant intervention in the context of addiction [41]. Studying the role of green spaces seems particularly valuable in rapidly urbanizing and highly urbanized countries in Northeast Asia where internet addiction is reported to be highly prevalent [42].

Additionally, it could be valuable for this line of research to add biological correlates of mental health to the outcome parameters for comparison with questionnaires and neurocognitive tests. Recently, hair cortisol analysis in children [43] and neuro-immunological measures [44] and mobile neuro-imaging [45] in adults were introduced in this field.

Furthermore, to shed light on other confounding and/or mediating variables of the established associations, future studies could try to build models assessing the separate roles of air and noise pollution, physical activity, social interaction and parental mental health. Longitudinal studies with broad screening tools adapted to different age groups are welcomed to investigate the impact on the developmental process and to raise the probability of causality. Attention should be paid to change of address during follow-up as mentioned earlier.

Finally, studies in rapidly urbanizing African and Asian countries seem to be very relevant for policy guidance.

## 5. Conclusions

In children and adolescents, there is significant evidence for an inverse relationship between green space exposure and emotional and behavioral problems. These beneficial associations are resistant to the adjustment for demographic and socio-economic factors and thus may represent an independent link. The presented evidence suggests potential partial mediation via physical activity, buffering of air pollution and social interaction. Future research is needed to shed more light on confounders, mediators, varying impacts during development and causality. The presented findings warrant the attention of policy makers, urban planners and health care workers in order to protect the mental health of children and adolescents in an urbanizing world, by providing sufficient exposure to green spaces.

## Figures and Tables

**Figure 1 ijerph-15-02668-f001:**
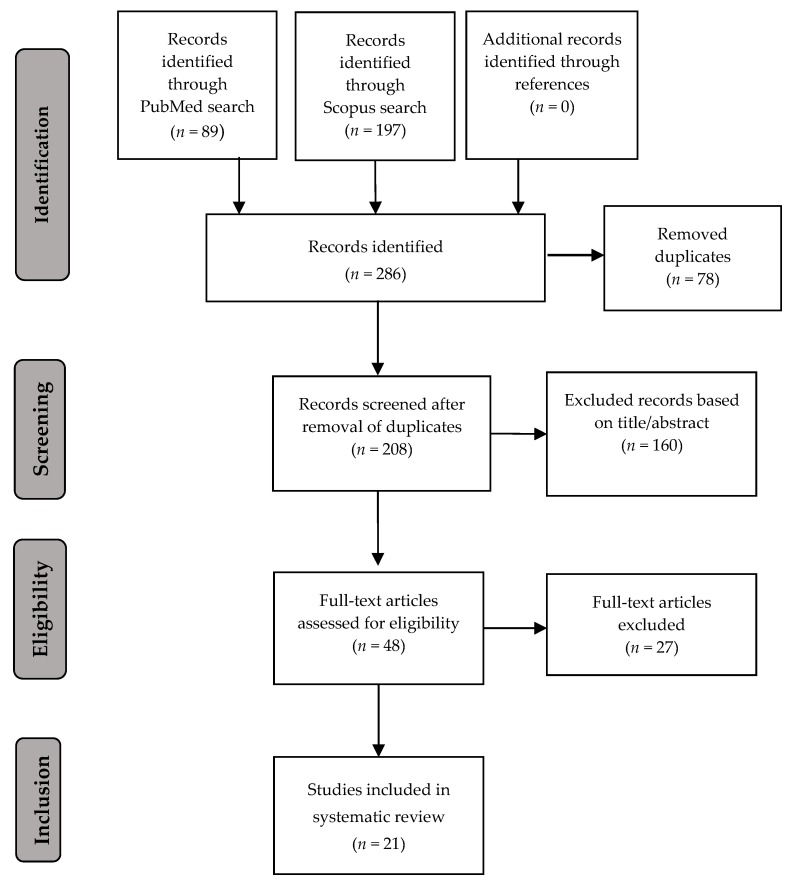
Search results. Figure based on Preferred Reporting Items for Systematic reviews and Meta-Analyses (PRISMA) guidelines by Moher et al. [11].

**Table 1 ijerph-15-02668-t001:** Study characteristics, methodology and results.

Author, Year, Country	Study Design	*N*, Age	Exposure Data Source	Exposure Area	Exposure Concept (Scale)	Outcome Instrument	Confounders Adjusted for in Model	Significant Results in Adjusted Model
Aggio et al., 2015, Scotland [15]	cross-sectional	3586 5–6 years	questionnaire	residence	walking time to nearest GS	SDQ	no adjusted model in publication	(Not adjusted) more than 20 min walking time to GS associated with higher scores on TDS.
Amoly et al., 2014, Spain [18]	cross-sectional	2111 7–10 years	NDVI, questionnaire	residence, school	average greenness (100 m, 250 m, 500 m); GS playing time; proximity to major GS	SDQ, DSM IV/ADHD	gender, school level, ethnicity, SES, parental marital status, nSES preterm birth, breastfeeding, environmental tobacco smoke, gestational maternal smoking, responding person	Average greenness inversely associated with TDS, H/I & DSM IV-ADHD score; green space playing time inversely associated with TDS, emotional and peer problems; proximity to major GS not associated with outcome parameters.
Balseviciene, 2014, Lithuania [19]	cross-sectional	1468 4–6 years	NDVI, land cover map	residence	average greenness (300 m); distance to city park	SDQ	age, gender, parenting stress, SES	Distance to city park positively associated with TDS, H/I, peer and conduct problems in low SES subgroup; no associations with average greenness.
Bezold et al., 2017, USA [20]	cross-sectional	9385 12–18 years	NDVI	residence	average greenness (250 m, 1250 m)	McKnight Risk Factor Survey	age, gender, ethnicity, grade level, SES, maternal history of depression, nSES, PM_2.5_	IQR increase in average greenness associated with 11% lower odds of high depressive symptoms.
Bezold et al., 2018, USA [21]	longitudinal	11,346 9–25 years	NDVI	residence	cumulative average greenness (1000 m)	Mc Knight Risk Factor Survey; CES-D	age, gender, ethnicity, SES, maternal history of depression, population density, nSES, PM_2.5_	IQR increase in cumulative residential greenness associated with 6% lower incidence of high depressive symptoms; stronger associations for under-18 year olds and in more densely populated areas.
Dadvand et al., 2015, Spain [22]	longitudinal	2593 7–10 years	NDVI	residence, school, commuting	average greenness (250 m residence, 50 m commuting route, 50 m school)	Compu-terized n-back, ANT	age, gender, SES, nSES, air pollution	Average greenness positively associated with 12 m development of two-back, three-back and ANT results.
Dadvand et al., 2017, Spain [23]	longitudinal	1875 4–7 years	NDVI, VCF	residence	cumulative average greenness (100 m, 300 m, 500 m), tree canopy cover	computerized K-CPT, ANT	age, gender, preterm birth, maternal cognitive performance, gestational smoking, environmental tobacco exposure, SES, nSES	Cumulative average greenness inversely associated with K-CPT omission errors and HRT-SE at 4–5 years, and with ANT HRT-SE at 7 years.
Dzhambov et al., 2018, Bulgaria [24]	cross-sectional	399 15–25 years	NDVI, SAVI, TCI, GIS, questionnaire	residence	average greenness (500 m), tree canopy cover, GS access/quality/usage	GHQ	age, gender, ethnicity, SES orientation of rooms duration of residence, time spent at home, air pollution, noise, population density	No direct associations; positive association via serial mediation (restoration, physical activity, social cohesion).
Feng and Astell-Burt, 2017, Australia [25]	longitudinal	4968 4–13 years	land cover map, question-naire	residence	amount of GS (SA2), GS quality	SDQ	age, gender, ethnicity, SES, nSES, urbanicity	Amount and quality of GS inversely associated with TDS, IS and ES, for all age groups; for older children GS quality more strongly inversely associatied with IS.
Feng, 2017, Australia [17]	cross-sectional	3083 12–13 years	land cover map, question-naire	residence	amount of GS (SA2), GS quality	SDQ	age, gender, SES, nSES, geographic remoteness	Amount of GS inversely associated with the parent-reported TDS and IS; GS quality inversely associated with both parent- and child-reported TDS, IS and ES; stronger associations found for parent-reported scores.
Flouri et al., 2014, UK [26]	longitudinal	6383 3–7 years	land cover map, question-naire	residence	amount GS (LSOA), use of GS	SDQ	age, gender, ethnicity, SES, family structure, use of GS, access to private garden, life adversity, nSES, maternal (mental) health, physical activity	Amount of GS inversely associated with emotional problems score for age 3–5 years.
Huynh et al., 2013, Canada [27]	cross-sectional	17,249 11–16 years	land cover map	school	amount GS (5000 m)	Cantrill ladder	age, gender, ethnicity, SES, nSES	Amount of GS not associated with well-being.
Kabisch et al., 2016, Germany [13]	ecological	30,427 5–6 years	land cover map	residence	amount GS (LEA) (/capita)	health visit	sub-district level: SES, ethnicity, measles immunization, participation in check-up, kindergarten attendance, tobacco exposure	Amount of GS inversely associated with deficits in visuo-motoric development.
Kim et al., 2016, USA [28]	cross-sectional	92 9–11 years	remote sensing data (NDVI-like)	residence	amount, number, size, distance to, cohesiveness of GS (400 m, 800 m)	PedsQL	age, gender, SES, nSES, BMI, physical activity	Larger and more tree areas positively associated with children’s health related quality of life.
Markevych et al., 2014, Germany [29]	cross-sectional	1932 10 years	land cover map, NDVI	residence	distance to nearest GS	SDQ	age, gender, SES, maternal age at birth, parental marital status, screen/outdoors time	Residence > 500 m away from nearest GS positively associated with TDS, peer relationship and H/I problems (after stratification; only association with H/I for boys), no associations for residential average greenness.
Richardson et al., 2017, Scotland [30]	longitudinal	2909 4–6 years	land cover map, question-naire	residence	amount GS and public parks (500 m), access to private garden	SDQ	age, gender, SES, parental mental health, nSES, hours of screen time	Private garden access strongly associated with TDS and H/I and to lesser extent with peer and conduct problems, neighborhood amount of GS associated with prosocial behavior scores, little evidence of influence on developmental trajectory.
Saw et al., 2015, Singapore [31]	cross-sectional	426 18–25 years	land cover map, question-naire	residence	distance from nearest GS, number of GS’s (1200 m), use of GS	LSS, Pos.and Neg. Affect Scale, PSS	age, gender, SES, physical activity, serious health problems, personality traits	Neither access to or use of GS associated with well-being.
Ward et al., 2016, New Zealand [14]	cross-sectional	72 11–14 years	geolocation and timing	all locations	time spent in GS	LSS; TDIW; HS; comp. CNS-Vital Signs test	age, gender, school, physical activity	Time spent in GS and physical activity positively associated with greater emotional wellbeing, no associations with neurocognitive development measures.
Wu et al., 2017, USA [12]	ecological	~3 × 10^6^ 5–12 years	land cover map	school district	amout forest and grassland (school district), amount (near-road) tree canopy.	prevalence autism	district level: ethnicity, gender, SES, road density	Amount of GS and tree cover metrics inversely associated with autism prevalence in high road density districts.
Younan et al., 2016, USA [32]	longitudinal	1287 9–18 years	NDVI	residence	average greenness (250 m, 350 m, 500 m, 1000 m)	CBCL-Agression	age, gender, ethnicity, SES, nSES, ambient temperature, traffic density and proximity to freeways and roads, maternal depression, gestational smoking	IQR increase in average greenness inversely associated with aggresive behaviour.
Zach et al., 2016, Germany [16]	cross-sectional	6206 6–12 years	questionnaire	residence	availablity of GS	SDQ	gender, country of birth, SES, single parenthood, crowding, traffic load	Non-accessibility of green space associated with TDS and hyperactivity/inattention problems.

Note: ADHD, Attention Deficit and Hyperactivity Disorder; ANT, Attention Network Task; CBCL, Child Behavior Checklist; CES-D, Center for Epidemiologic Studies—Depression Scale; comp. CNS-Vital Signs test, computerized Central Nervous System-Vital Signs test; DSM IV, Diagnostic and Statistical Manual Fourth Edition; ES, Externalizing Scale; GHQ, General Health Questionnaire; GIS, Geographical Information System; GS, Green Space; H/I, hyperactivity/inattention; HRT-SE, Hit Reaction Time Standard Error; HS, Happiness with life as whole Scale; IQR, Interquartile Range; IS, Internalizing Scale; LEA, Living Environment Area; LSOA, Lower layer Super Output Area; LSS, Life Satisfaction Scale; K-CPT, Kiddies Continuous Performance Task; NDVI, Normalized Difference Vegetation Index; nSES, neigborhood Socio-Economic Status; PedsQL, Pediatric Quality of Life scale; PM_2.5_, Particulate Matter > 2.5 µm; PRS, Perceived Restorativeness Scale; SA2, Statistical Area 2; SAVI, Soil-Adjusted Vegetation Index; SDQ, Strengths and Difficulties Questionnaire; SES, Socio-Economic Status (i.e., family income, parental education and/or parental employment); TCI, Tree Cover Index; TDIW, Ten Domain Index of Happiness; TDS, Total Difficulties Scale; VCF, Vegetation Continuous Fields.

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
