# Peer review of "Impact of Green Space Exposure on Children’s and Adolescents’ Mental Health: A Systematic Review"

_ijerph, 2018, doi:10.3390/ijerph15122668_

Round 1
Reviewer 1 Report
This is a very well structured and comprehensive, well-written review. I don’t have many questions or comments. Perhaps one:
Lines 80-83
While I feel that keywords related to mental health and neurocognitive development ("mental health", "well-being", development*) and keywords related to childhood and adolescence (child*, 83 adolescen*, infan*, “young adult*”, youth) are quite well chosen I’m not so sure about it in the case of keywords relating to green space ("green space", greenspace, greenness). Shouldn’t there be more variety in the choice here? I would rather see ke words such as: green space, greenery, park, garden or even nature (in some cases authors might relate to urban greenery as “natural environments”)
You are analysing studies that were taking place in school environments - isn’t it possible that in a case the greenspace adjacent to a school, authors would use keywords such as gardens or natural playgrounds rather than greenspace?
I know this is further discussed in “4.7. Strengths and Limitations” but perhaps it would be beneficial to the study to show how many “hits” (results) came off each search (each keyword or keyword combinations). As I feel that words such as greenness are not very popular in the scientific / landscape architecture language. In a next step you could eliminate such keywords, replacing them possibly with different ones (greenness >> greenery or park).
Author Response
Please see PDF 'response review 1' in attachment.

Reviewer 2 Report
Thank you for inviting me to review the paper on "Impact of Green Space Exposure on Children’s and Adolescents’ Mental Health: a Systematic Review". This is an important paper to highlight the inverse hypothesis, i.e. the adverse mental health of parents reduce green space exposure on their children. This review is deserved to be published in IJERPH. I have the following recommendations to enhance the quality of this important review.1. Under limitations, 416-417, the authors mentioned about low and middle-income countries. The authors should also state that almost all studies included in this systematic review were from western countries and did not include studies from Northern Asia including China, Korea and Japan. Children and adolescents in these three countries faced a unique problem which is internet addiction (Please refer to this paper: https://www.ncbi.nlm.nih.gov/pubmed/25405785). The other limitation of this study is that mental health is narrowly defined as emotional well being and quality of life but forget about other problems like addiction. Exposure to the green environment can reduce addiction (Reference: https://www.tandfonline.com/doi/abs/10.1300/J076v29n03_11). Please discuss the above points and include these references under limitation.
2. Under 4.8, the authors suggested questionnaire based measurement such as SDQ. The authors need to acknowledge that the questionnaire is associated with self-report bias and can lead to inaccurate findings. The authors should discuss the latest trend of measuring changes in neuroinflammatory function
(Reference: https://www.ncbi.nlm.nih.gov/pubmed/30096932) and portable neuroimaging (Reference: https://www.ncbi.nlm.nih.gov/pubmed/?term=29060418) when assessing participants' response in green space. These novel measurement methods can be applied to young people. Please include these statements in future research direction.
Author Response
Response to review 2
The suggestions made by reviewer two regarding the lack of studies in Northern Asian countries, internet addiction and the correlation between these geographical regions and this mental health problem are added to the manuscript in the discussion section. The notion of possible self-report bias as a limitation of questionnaires is equally discussed in the renewed version of the manuscript. In the future research suggestions we have added the suggestion of taking into account biological correlates of mental health such as neuro-inflammatory and neuro-imaging measures. We consider these remarks as valuable contributions to this review.